# Examination of VOC Concentration of Aroma Essential Oils and Their Major VOCs Diffused in Room Air

**DOI:** 10.3390/ijerph19052904

**Published:** 2022-03-02

**Authors:** Toshio Itoh, Yoshitake Masuda, Ichiro Matsubara, Junichirou Arai, Woosuck Shin

**Affiliations:** 1National Institute of Advanced Industrial Science and Technology (AIST), Nagoya 463-8560, Japan; itoh-toshio@aist.go.jp (T.I.); masuda-y@aist.go.jp (Y.M.); matsubara-i@aist.go.jp (I.M.); 2Technology and Innovation Center, Daikin Industries Ltd., Nishi-Hitotsuya, Osaka 566-8585, Japan; junichirou.arai@daikin.co.jp

**Keywords:** essential oil, aroma oil, melissa, lavender, tea tree, eucalyptus, GC/MS

## Abstract

This study analyzed temporal variation of the composition of volatile organic compounds (VOCs) at different diffusion time of gaseous phase of aroma compounds of four essential oils, lavender, tea tree, eucalyptus, and melissa. GC/MS methodology with the trace gas sampling by a thermal desorption tube is used to quantitatively determine the concentration of the corresponding 14 kinds of major and original VOCs in four essential oils. This study revealed for the first time that the concentration level of gaseous phase composition is varied, with a diffusion time from that of the liquid phase at equilibrium with it and the VOCs in the essential oils are classified into two groups, depending on whether their concentration with the time. It is verified that the total concentration of VOCs of these essential oils in the room air diffused by the ultrasonic diffuser is as low as 0.6 ppb and decreased soon below 0.1 ppb.

## 1. Introduction

Aromatherapy is a very popular complementary therapy and is widely used in many health care settings and services. For example, the effect of the treatment of migraine headache and sleep disturbances by lavender oil [1,2], suppression of the inflammatory mediator by tea tree oil [3], anti-inflammatory activity on bronchial asthma by eucalyptus oil [4], and management of agitation in dementia by melissa oil [5]. Some of the people believe that their routine healthcare on aromatherapy is low cost for preventing to take medical care for illness. However, there are very few medical or scientific publications on aromatherapy [6], and there is no guidance available about the introduction and use of aromatherapy in routine healthcare.

There are many reports and demonstrations on the effect of the aroma ingredient by clinical studies [7]. However, for the diffusion of aroma and the concentration of its composition in room air, no quantitative analysis has been reported up to date, and monitoring technology of the volatile organic compounds (VOCs) in aroma essential oil is not yet developed. The difficulties are that the effects of aromatherapy under investigation are not easy to measure, and their concentration in room air seems likely to be very low. Moreover, a sound rationale should be provided of the effect of the aroma ingredient for the use of aromatherapy as a medical intervention [8,9]. In the absence of clear efficacy data for well-being and other health effects, it is probably considered as alternative medicine of little scientific merit, such as homeopathy.

Before designing clinical study and experiments on aromatherapy, it is necessary to prepare a systematically detailed clinical strategy plan and to estimate how much the volunteer inhale the VOC of aroma in room air in advance. Therefore, it is necessary to lower the hurdles for conducting clinical trials by conducting preliminary experiments to clarify the behavior of the airborne concentrations of essential oils and effective ingredients in the indoor space and their fluctuations over time. These data for fragrance materials should be obtained to make a valid assessment incorporated with the current state of health and environmental risk assessment methods [10]. 

In this study, gas chromatography mass spectroscopy (GC/MS) odor analysis has been performed to clarify the concentration of the ingredients of essential oils in room air and their behavior over diffusion time. Essential oils are diffused using an ultrasonic diffuser, which is a common and the easiest way to diffuse out the VOCs into the room air, thus as to improve human health for a period of daily activity or for a time of sleep. This simple device can offer a safer alternative to oral intake, especially if a person wants to take the ingredients of the oil in a controlled manner. This study of GC/MS analysis of natural essential oils and VOCs in these oils was conducted as a preliminary test before the clinical study. This survey of the airborne concentrations of essential oils and effective components in the indoor space and the behavior of fluctuations over time will be used as quantitative information in the clinical study, where human volunteers inhale these compounds.

## 2. Materials and Methods

### 2.1. Essential Oils and Major Components

Essential oils: Four types of essential oils (scientific name and vender), lavender (Lavandula angustifolia, Ryohin Keikaku Co., Ltd., Tokyo, Japan), tea tree (Melaleuca alternifolia, Ryohin Keikaku Co., Ltd., Tokyo, Japan), eucalyptus (Eucalyptus, Ryohin Keikaku Co., Ltd., Tokyo, Japan), and melissa (Melissa officinalis, Insent Co., Nagano, Japan), were used in this study. Major components in the essential oils are researched from a text of phytotherapy [11] and reference papers and internet websites [12,13,14,15,16,17,18,19], as listed in Table 1. 

The components and their ratios searched by two different resources (excluding melissa, who has no component ratio in internet surveys) are roughly consistent, and the small deviations are inevitable because the essential oils are naturally derived and can fluctuate depending on the production area and production time. The ratio for melissa and eucalyptus was adopted by phytotherapy, and lavender and Tea-tree by internet survey.

Effective components: 14 kinds of components were selected from the major components of four essential oils as effective components in this study. They are selected as representative aromatic constituents of four essential oils by excluding some major constituents which is considered closer to solvent rather than aromatic component, such as terpinolene, geraniol. Twelve effective components except for geranial and neral were used, and citral was used instead of geranial and neral because citral is a mixture of geranial and neral. Figure 1 shows the structural formulae of the 14 kinds of effective components. As most of both essential oils and chemicals are supplied as mixtures of enantiomers [20], also in this study, enantiomeric classification is simplified and omitted. The components are indicated by the numbers of shoulder brackets in Table 1 and Figure 1. 

### 2.2. Diffusion of Essential Oil and VOC into Room Air

In the office room, the essential oils mentioned above were diffused using a commercially available aroma diffuser (Ryohin Keikaku Co., Tokyo, Japan). This aroma diffuser is selected from the viewpoint that it is desirable to use a diffuser that is easily available and has low peculiarities. It has a tank capacity of 350 mL for distilled water and possesses diffusing water on around 100 mL in 60 min. Essential oils of 0.5 mL are dropped with a microsyringe onto distilled water at the beginning, as shown in Figure 2. The aroma diffuser uses ultrasonic waves to diffuse water and the essential oils in the tank to produce a fragrant mist. 

The room air samples were collected with thermal desorption (TD) tubes (GERSTEL, GERSTEL K.K. Japan, Tokyo, Japan) as shown in Figure 3a, from the distance of 500 mm from the aroma diffuser while diffusing in the office room. The volume of collected air was 1 L which was pumped with a rate of 200 mL/min for 5 min. The interval of the collection is as shown in Figure 4. The 5-min-collection was carried out immediately on the start of aroma diffuser diffusing, hereinafter referred to as 0 min. In the same manner, the 5 min collection was also carried out from 30 to 35 min and from 60 to 65 min, hereinafter referred to as 30 and 60 min, respectively.

The effective components are diffused in the air on the same method as above. However, in the office room, diffusing chemicals should be avoided from a safety point of view. The diffusion in the air was carried out in a draft chamber of 650 × 1300 × 1250 mm^3^, as shown in Figure 3b. During the diffusion, the height of a shutter was controlled for controlling flow rate of ventilation at 0.5 m/s.

### 2.3. GC/MS Analysis

Analysis of diffused essential oils and effective components in TD tube is carried out by GC/MS (Agilent GC 6890 and MS5973, Agilent Technologies Japan Ltd., Tokyo, Japan) with autosampler for TD tubes (GERSTEL TDS A2, GERSTEL), as shown in Figure 5. Collecting volatile compounds using TD tube is a highly sensitive GC/MS-analyzing method because of large volume collection thus it is more efficient and sensitive than other extraction methods such as solid-phase microextraction (SPMI) for trace-level analysis. Thermal desorption is the process of collection and desorption of analytes from solid sorbents using heat and a flow of inert gas, and then the analytes are trapped in a cold trap prior to entering the analytical column, resulting in narrow and symmetric peaks. The condition of GC/MS is listed in Table 2. The column used in this study, DB-WAX, is a high polarity type made of Polyethylene glycol (PEG) and recommended type for food, fragrance, and flavor applications.

The calibration method of GC/MS peak intensity and component concentration is as below. One microliter of essential oil was injected into 10 L of pure air filled in PDVF (polyvinyridene fluoride) bag, and the bag was kept overnight. The concentration of each component, linalool γ-terpinene, 1,8-cineole, and caryophyllene, for lavender, tea tree, eucalyptus, and melissa, respectively, in the bag can be calculated from the component ratio contained in the essential oil. Connecting the bags with TD tubes and collecting 100 mL, then the TD tubes were analyzed by the GC/MS system, and obtaining the peak area of each component was used as the reference. The relationship between concentration and peak area of each component is calculated. 

## 3. Results

### 3.1. GC/MS Results of Essential Oils Diffused into Office Room

Figure 6 shows the GC/MS chromatogram of lavender oil collected in the office room as an example. Descriptions of the peaks were from the analysis results using the MS database. The chromatogram of room air without any essential oils and effective components is the chromatogram [bg] in Figure 6, indicating that the main components of lavender oil were not included in the ingredients that were always present in the room. As for the chromatogram of the room air samples of the lapsed time of 0, 30, and 60 min from the oil diffusion test, [0 min] [30 min] [60 min], these peaks were compared to that of the sample gas in the PVDF bag as a reference, [bag], for obtaining the concentration of each component.

The concentration of each component was calculated and listed in Table 3, with the molecular formula, molecular weight, and retention time. For the components which were unknown or chemicals that originated from the extraction process, such as ethanol and benzene, these concentrations were calibrated with the calculation based on the peak area of linalool.

For other essential oils, tea tree, eucalyptus, and melissa were also carried out on the same analysis and listed in Table 4, Table 5 and Table 6, respectively. GC/MS chromatogram of three essential oils are found in Appendix A, Appendix A. 

The concentration level of each main component of essential oils was very low and several to several tens of ppt (parts per trillion). The GC/MS peaks were calibrated with the calculation based on the peak area of linalool. For tea tree oil, Eucalyptus oil, and melissa oil, the peak area of γ-terpinene 1,8-cineole, and caryophyllene, was used for the calibration, respectively. 

For lavender oil, see Figure 7, the concentration level of each main component, listed in Table 1, was several to several tens of ppt, and the total concentration of main components was up to 75 ppt. β-Phellandrene, α-terpineol, camphor, and lavendulol were not detected in this study. An interesting feature found in this study was that the concentration of each component varied with the elapsed diffusion time. That is, the ratio of components changed over diffusion time. However, the total concentration decreased slowly. This can be related to the molecular weight or retention time of GC/MS system. Four components of 1,8-cineole, trans-beta-ocimene, cis-beta-ocimene, and 3-octanone, whose GC/MS peaks appeared before the retention time of 14.2 min, show high concentration at 0 min then decreased fast by the elapsed time of 30 min. Other components, linalool, linalyl acetate, terpine-4-ol, and lavandulol acetate, whose GC/MS peaks appeared after the retention time of 18.5 min, showed high concentration at an elapsed time of 30 min. 

For tea tree oil, see Figure 8, the concentration level of each component was several to several tens of ppt, similar to the case of lavender. However, the total concentration at elapsed time 0 was as high as 272 ppt and decreased to 30 ppt by 30 min. α-Terpinenol was detected from PVDF bag as a reference, however, not detected from room air. A total of 9 components with a retention time of up to 14.70 min showed high concentration at the beginning. Especially the concentration of γ-terpinene was high as 104 ppt at 0 min. After the retention time of 19.58 min, terpinen-4-ol delayed and showed peak concentration at 30 min. 

For Eucalyptus oil, see Figure 9, the concentration level of each main component, listed in Table 1, was higher than that of other essential oils. Specifically, the concentration of 1,8-cineolewere 375 ppt at 0 min. (E)-pinocarveol and globulol were not detected in this study. Other ingredients started at tens of ppt. The total concentration at 555 ppt at the start at 1/6 in 30 min and 1/20 in 60 min. The concentration of any component decreases over time. All four components of the detection were detected before the retention time of 14.70 min, which was the boundary line where the peak of lavender and tea tree concentrations was either [0 min] or [30 min].

For Melissa oil, see Figure 10, the concentration of each component was several to tens of ppt, and the total concentration was about 17 to 55 ppt. Germacrene D, caryophyllene oxide, and geraniol were not detected in this study. In any component, the peak concentration appeared late [30 min]. All four components of detection were detected after the retention time 14.70 min, which is the boundary line where the peak of lavender and tea tree concentrations was either [0 min] or [30 min].

### 3.2. GC/MS Results of 14 Effective Components Diffused into Draft Chamber

Effective components, which are major VOCs in the four essential oils, are diffused into the draft chamber, and the temporal changes of their concentration are measured by GC-MS. For safety reasons, the test of VOC diffusion was carried out in the draft chamber, but other experimental conditions were the same as that of essential oils. The details of GC/MS chromatogram of these components are found in Appendix A, in Appendix A.

Figure 11 shows the results of the concentration of the components in 0, 30, and 60 min. The component whose concentration decreased with the elapsed time shows a concentration of several parts per billion (ppb) immediately after the start of spraying but decreased to 0.2 ppb or less after 30 min. The tendency confirmed in this study that the higher the concentration at 0 min, the lower the concentration tended to be at 30 min was shown again clearly, and especially for α-terpinene, its GC/MS peak was not detected at 30 min. The components of the delayed or high molecular components, their concentration level was less than 1 ppb at 0 min, but the concentration was not so much changed. A specific change is shown in the cases of linalyl acetate and terpinen-4-ol. Their concentration in the room air gradually increased. The concentrations of linalool and α-terpineol reached their maximum at 30 min.

Before and after the retention time band from 14.6 to 17.7 min of the GC/MS, the components can be divided into two groups; with one group decreasing its concentration with time, including α-pinene, β-pinene, α-terpinene, (+)-limonene, 1,8-cineole, γ-terpinene, and p-cymene; and the other group of a delayed peak of concentration with the time, including citronellal, linalool, linalyl acetate, terpinen-4-ol, neral, α-terpineol, and geranial. This tendency was similar to the behavior of the concentration change of each component in the test in which the essential oil was diffused.

## 4. Discussions

### 4.1. Temporal Change of the Concentration of VOCs in Essential Oils

The above results are summarized in Table 7. The component ratio fluctuates with elapsed time. For the reason why the scent changes over time, the peak concentration differs depending on the component. That is, eucalyptus and tea tree showed a dramatic reduction by elapsed time, but melissa and lavender lasted for an hour.

The aroma of fragrances is also classified into three basic categories [21], so-called notes, are similar to how musical notes make up a song, top notes, heart notes, and base notes. Top notes have higher volatility, and they appear faster within 10 or 15 min, while base notes are longer-lasting and appear for hours. This categorization or note is strongly related to how volatile the note is. Among four oils, lavender is only one top-note aroma, but the concentration was constantly lasted.

An almost linear relationship between the molecular weight and the retention time is observed as listed in Table 3, and several order mismatches are due to the polarity of the component, as the column material used in this study, DB-WAX, is the high polarity of PEG. Since components with a long retention time tend to have a large molecular weight, those with low volatility are expected to be applicable.

An interesting feature of the case of tea tree oil, the most lightweight component showed an explicate decrease with time, but terpinen-4-ol, which is considered the main component of the oil, kept a relatively high concentration for a long time. The content of is known as 38%, see Table 1, but was found to be small, around 5% in this study, and the main component was γ-terpinene, whose concentration decreased fast with diffusion time. In contrast, 1,8-cineole, which is the main component 80% in eucalyptus, showed high concentration and decreased gently to be 30 ppt after 60 min diffusion.

These reasons are that components having polar functional groups, such as hydroxyl groups and aldehyde groups, are less volatile and more soluble in water than components without these functional groups. In this study, 350 mL of water with 0.5 mL of essential oils was diffused by ultrasonication. Insoluble components in water are spread on the surface of the water and should be easily diffused immediately, whereas soluble components are dissolved in water, and the concentration is low, but long-term diffusion can be expected. That is, α-pinene, β-pinene, α-terpinene, (+)-limonene, 1,8-cineole, γ-terpinene, and p-cymene included in the one group of decreasing its concentration with the time in Figure 11 do not have the polar functional groups (excluding ether group), whereas citronellal, linalool, linalyl acetate, terpinen-4-ol, neral, α-terpineol, and geranial included in the other group of a delayed peak of concentration with the time have polar functional groups. The same tendency can be seen as the diffusion of 14 effective components in Figure 11 with respect to the analysis results of the essential oils in Figure 7, Figure 8, Figure 9 and Figure 10.

### 4.2. Safety Issue of VOCs

One important purpose of this study is to measure the concentration level of VOCs or the component of essential oils diffused into the room air by the common use of diffusion. When the aroma is used, the safety issue is a critical consideration, for example, to understand the concept of the Threshold of Toxicological Concern [22], and a question is whether the exposure of VOC is below the critical threshold for health or not. For the concentration of VOC is defined by ISO 16000-X series, and WHO and or Japanese government [23] declared the regulation on the maximum level of VOCs in the room air, for example, 260 μg/m^3^ or 6.9 ppb for toluene, 100 μg/m³ or 8.15 ppb for formaldehyde. There are several Japanese codes related to the use of fragrance or aroma, such as Food Sanitation Law, Chemical Substances Control Law, Pharmaceutical Affairs Law, but no clear threshold for aromatic VOC can be found. For fragrances, International Fragrance Association (IFRA) regulation or standard is only one reference regarding the acquisition, development, manufacture, and sale of raw materials for fragrances, which is checked by the expert agency of Research Institute for Fragrance Materials, Inc. (RIFM) for safety and health [24].

An important fact we have confirmed for the first time is that the total concentration of VOCs in the room air diffused by the ultrasonic diffuser is as low as 0.6 ppb and decreased soon below 0.1 ppb, as shown in Figure 7, Figure 8, Figure 9 and Figure 10. Considering the threshold level and safety guidance discussed above, the concentration of VOCs in room air diffused by a common ultrasonic diffuser analyzed in this study is safe even for repeated use. For the VOC exposure, a more detailed study on inhalation is required to conduct a safety assessment. A 2-box air dispersion model [25] is a good example for indoor environmental air, which determines air exposure concentrations in two connected, enclosed zones. This information can be combined with our results in this study and be meaningful to design a clinical study with aroma compounds of four essential oil.

## 5. Conclusions

For the use of aromatherapy, it is necessary to evaluate the amount of volatile chemicals in room air that the volunteer inhales. In this study, GC/MS odor analysis was performed to clarify the concentration of the ingredients of essential oils in room air and their behavior over diffusion time using the common ultrasonic diffuser.

GC/MS analysis with the trace gas sampling by thermal desorption tube verified the concentration of the VOCs in four essential oils, lavender, tea tree, eucalyptus, and Melissa, diffused by ultrasonic diffuser into room air and their behavior over diffusion time. At the same time, the major and original 14 VOCs in the essential oils are also diffused by an ultrasonic diffuser, and their concentration is evaluated.

The concentration of the corresponding 14 kinds of major and original VOCs in the essential oils varied with diffusion time, and the VOCs can be classified into two groups, one group is of decreasing its concentration with the time and the other group of a delayed peak of concentration with the elapsed time.

It is verified that the total concentration of VOCs of these essential oils in the room air diffused by the ultrasonic diffuser is as low as 0.6 ppb and decreased soon below 0.1 ppb. We can suggest that the concentration level and its temporal changes of VOCs and the oils confirmed in this study is very informative and critical when we work to create any guideline used for the clinical studies on how the VOCs and essential oils affect the health of humans.

## Figures and Tables

**Figure 1 ijerph-19-02904-f001:**
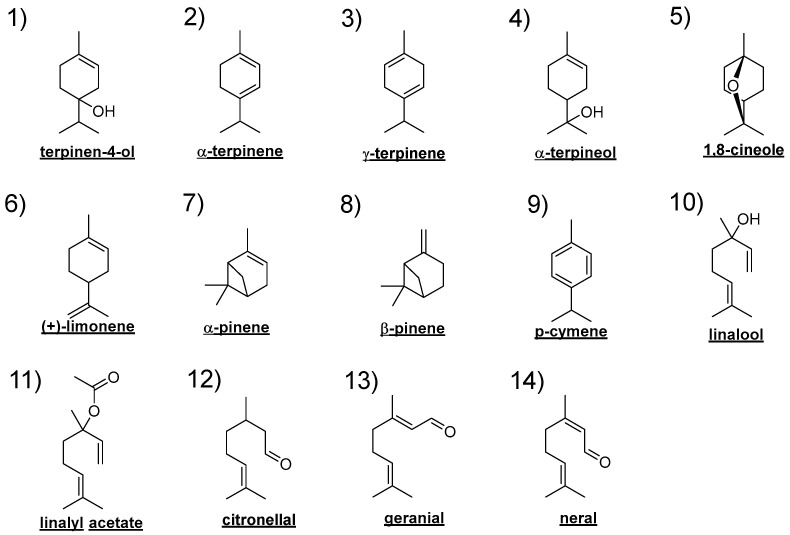
Structural formulae of 14 effective components in this study. (**1**) terpinene-4-ol, (**2**) α-terpinene, (**3**) γ-terpinene, (**4**) α-terpineol, (**5**) 1,8-cineole, (**6**) (+)-limonene, (**7**) α-pinene, (**8**) β-pinene, (**9**) p-cymene, (**10**) linalool, (**11**) linalyl acetate, (**12**) citronellal, (**13**) geranial, (**14**) neral.

**Figure 2 ijerph-19-02904-f002:**
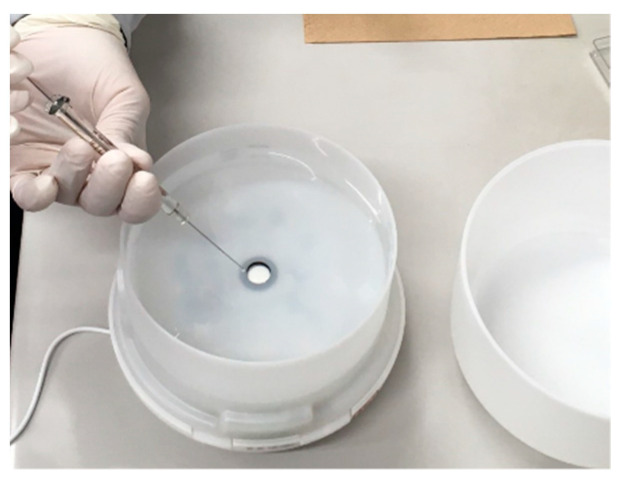
Preparation of diffusing and collecting test: Dropwise 0.5 mL of essential oils and effective component to 350 mL water in the aroma diffuser.

**Figure 3 ijerph-19-02904-f003:**
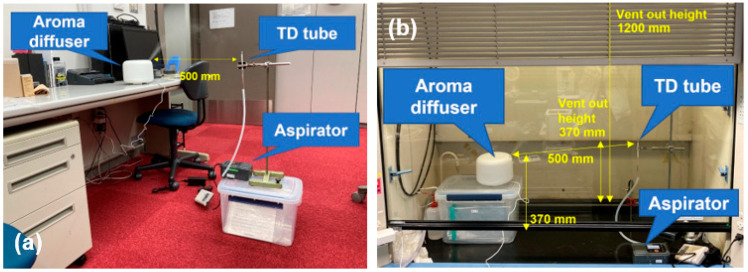
(**a**) Diffusion of essential oils in the office room by aroma diffuser and (**b**) diffusion of 14 effective components in the draft chamber. The air in the office room and the draft chamber is collected by thermal desorption (TD) tubes using aspirator.

**Figure 4 ijerph-19-02904-f004:**
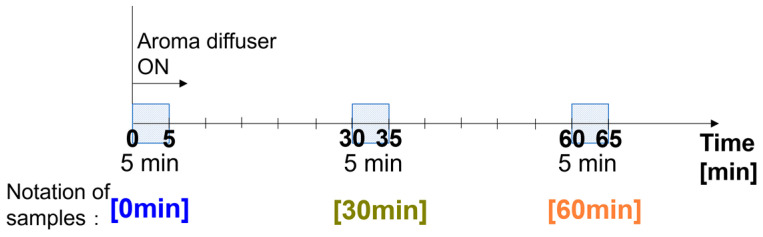
Schedule of collecting air, including diffused essential oils and effective components by TD tubes.

**Figure 5 ijerph-19-02904-f005:**
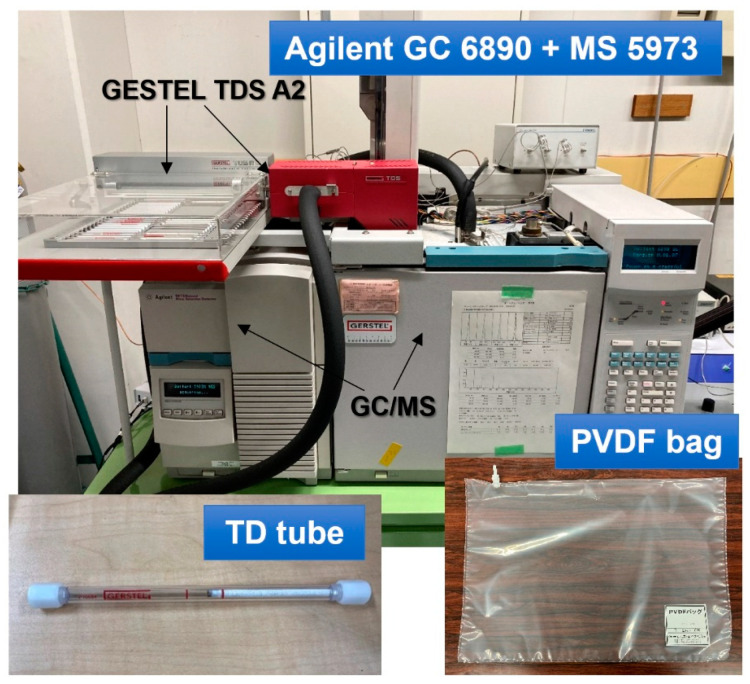
GC/MS system, TD tube, and PVDF bag.

**Figure 6 ijerph-19-02904-f006:**
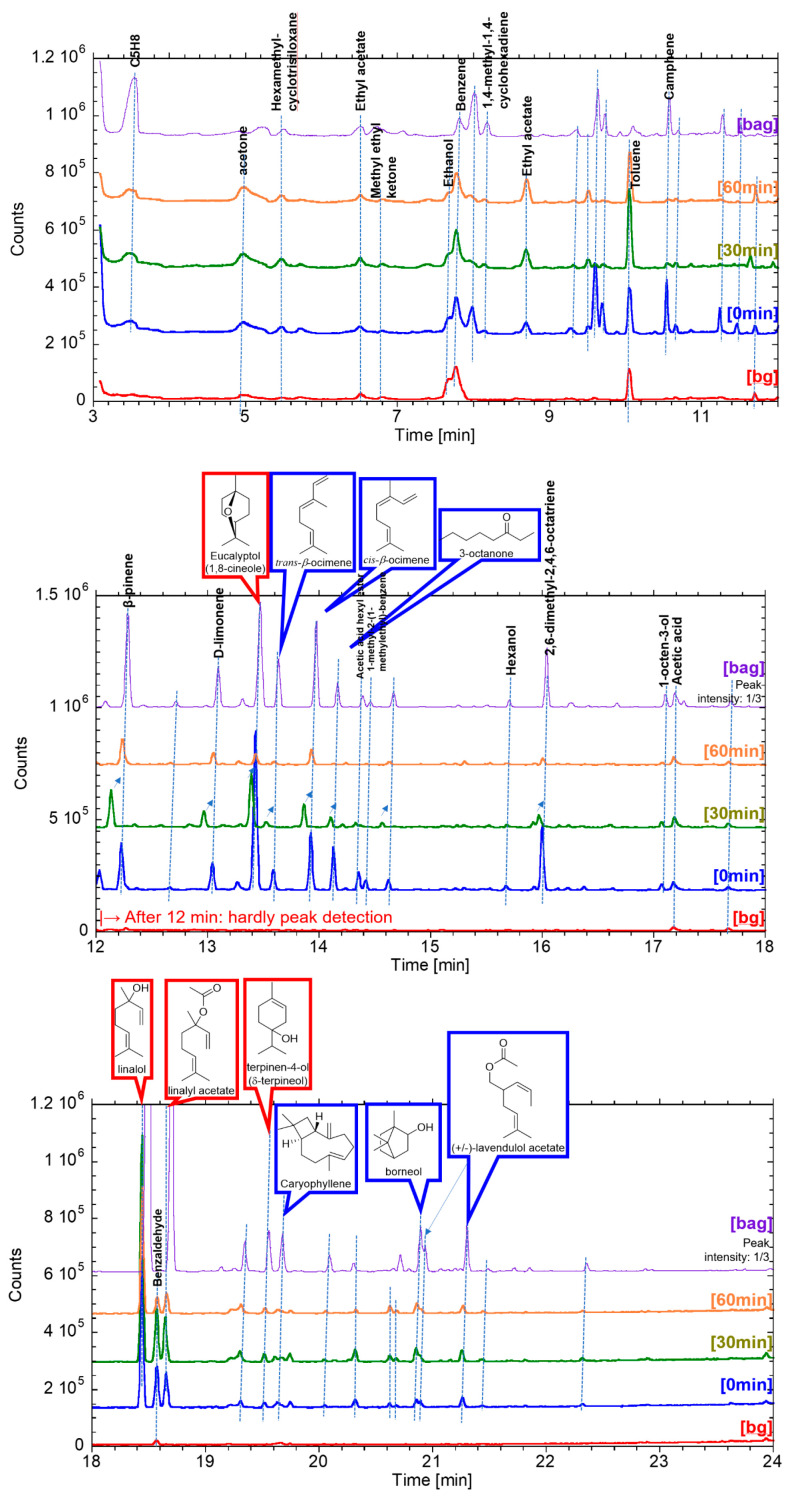
GC/MS chromatogram of lavender. Descriptions of the peaks were from the analysis results using the MS database, and descriptions with structural formulae mean major components of lavender, as listed in Table 1. Samples of [bg] and [bag] chromatogram were the room air before lavender-diffusing as a background measurement and gas from PVDF bag as a reference, respectively.

**Figure 7 ijerph-19-02904-f007:**
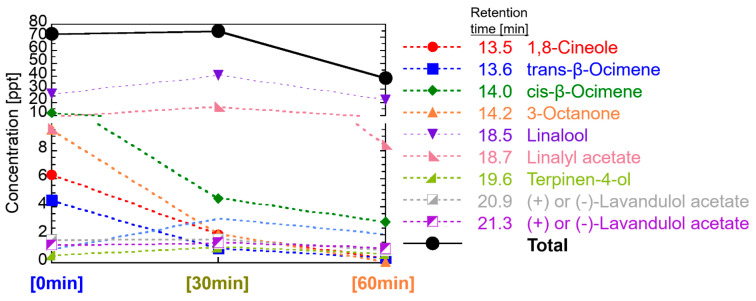
Temporal change of VOCs in the room air diffused by lavender oil.

**Figure 8 ijerph-19-02904-f008:**
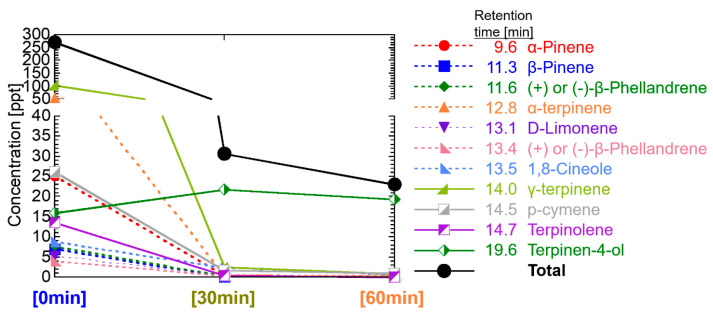
Temporal change of VOCs in the room air diffused by Tea tree oil.

**Figure 9 ijerph-19-02904-f009:**
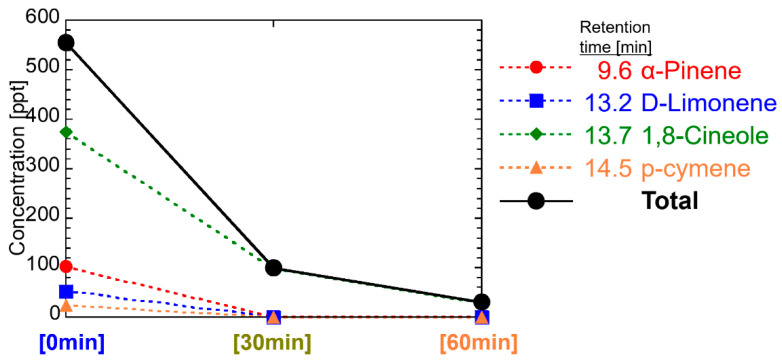
Temporal change of VOCs in the room air diffused by Eucalyptus oil.

**Figure 10 ijerph-19-02904-f010:**
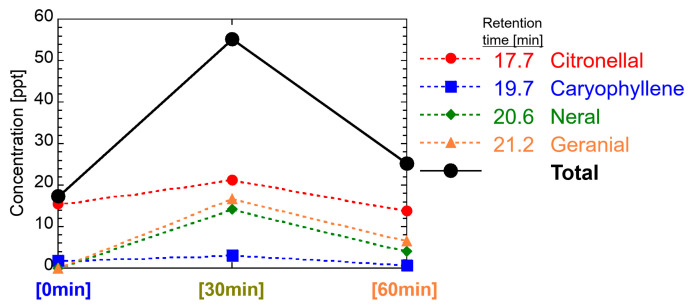
Temporal change of VOCs in the room air diffused by Melissa oil.

**Figure 11 ijerph-19-02904-f011:**
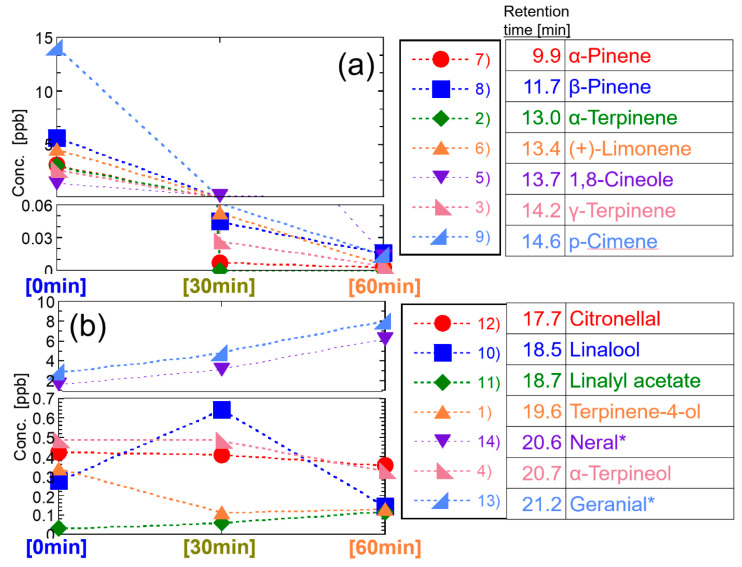
Summary of the temporal changes of 14 effective components diffused in the room air. (**a**) before, (**b**) after the retention time around 16 min. * Calculated by assuming the molar ratio of neral and geranial contained in citral is 1:1.

**Table 1 ijerph-19-02904-t001:** The concentration of major components in four essential oils researched from the book, references including internet websites.

	Reference Book [11]		References and Internet Websites [12,13,14,15,16,17,18,19]	
Melissa	(13) Geranial 25.4% (14) Neral 17.9% β-caryophyllene 9.7% (12) Citronellal 8.9%	Germacrene D 6.5% Caryophyllene oxide 5.4% Geraniol 4.6%	(13) Geranial 45.2% (14) Neral 32.8% (12) Citronellal 8.7% β-caryophyllene 3.3%	Caryophyllene oxide 2.1% β-Pinene oxide 1.1% Germacrene D below 0.05%.
Lavender	(10) Linalool 44.4% (11) Linalyl acetate 41.6% Lavendulyl acetate 3.7%	β-caryophyllene 1.8% (1) Terpinen-4-ol 1.5% Borneol 1.0%	(11) Linalyl acetate 38.0% (10) Linalool 31.9% cis-β-ocimene 4.0% trans-β-ocimene 2.5% (1) Terpinen-4-ol 2.2% Lavendulyl acetate 2.3%	3-octanone 1.1% (5) 1,8-cineole + β-phellandrene 1.0% (4) α-terpineol 2.8% Camphor 0.46% Lavendulol 0.42%
Tea tree	(1) Terpinen-4-ol 39.8% (3) γ-terpinene 20.1% (2) α-terpinene 9.6% Terpinolene 3.5%	(5) 1,8-cineole 3.1% (4) α-terpineol 2.8% (9) p-cymene 2.7% (7) α-pinene 2.4%	(1) Terpinen-4-ol 37.56% (3) γ-terpinene 22.27% (2) α-terpinene 11.43% (9) p-cymene 4.00% Terpinolene 3.75% (7) α-pinene 2.63%	(5) 1,8-cineole 2.21% (4) α-terpineol 2.01% β-phellandrene 0.97% (6) (+)-limonene 0.96% (8) β-pinene 0.78%
Eucalyptus	(5) 1,8-cineole 74.7% (7) α-pinene 9.2% (6) (+)-limonene 5.4%	(E)-pinocarveol 3.4% Globulol 2.7% (9) p-cymene 2.4%	(Globulus) (5) 1,8-cineole 70–90% (7) α-pinene 10–20% (6) (+)-limonene below 10%	(Radiater) (5) 1,8-cineole 60–70% (7) α-pinene 2–5% (6) (+)-limonene 5–10% (Citriodoura) (12) Citronellal 65–75%

**Table 2 ijerph-19-02904-t002:** Conditions of GC/MS.

Instrument	Agilent GC 6890 + MS 5973N
Column	DB-WAX UI; length: 30 m, diameter: 0.25 mm, film: 0.50 μm
Column oven	hold at 40 °C for 5 min heat up to 240 °C at a rate of 10 °C/min hold at 240 °C for 10 min
MS scan	m/z: 27–450
Wait time [min]	3
Threshold	100
Scan [s]	3.4
Scan velocity [μ/s]	1.562 (N = 2)
Cycle time	293.36
Step size [m/z]	0.1

**Table 3 ijerph-19-02904-t003:** The concentration of each component from room air including diffused lavender essential oil.

Retention Time [min]	Components	Molecular Formula	Molecular Weight	Concentration [ppt] *^5^		
				0 min	30 min	60 min
6.74	Ethyl acetate *^3^ *^4^	C_4_H_8_O_2_	92.8	4.4	6.1	4.1
7.68	Ethanol *^3^ *^4^	C_2_H_6_O	98.1	14	15	11
7.80	Benzene *^3^ *^4^	C_6_H_6_	95.9	16	16	15
8.01	Unknown *^3^	-	-	4.7	4.6	3.8
9.36	Unknown *^3^	-	-	2.7	0.36	-
9.63	Unknown *^3^	-	-	23	2.5	-
9.72	α-Phellandrene *^3^	C_10_H_16_	136.1	7.2	0.76	-
10.09	Toluene *^3^ *^4^	C_7_H_8_	98.0	15	25	15
10.57	Camphene *^3^	C_10_H_16_	136.1	12	1.1	0.55
12.28	β-Pinene *^1^	C_10_H_16_	136.1	13	10	7.3
13.10	D-Limonene *^1^	C_10_H_16_	136.1	6.5	4.2	3.0
13.47	1,8-Cineole *^1^ *^2^	C_10_H_18_O	154.1	6.3	2.0	0.33
13.64	trans-β-Ocimene *^2^	C_10_H_16_	136.1	4.5	1.0	0.37
13.97	cis-β-Ocimene *^2^	C_10_H_16_	136.1	12	4.6	2.9
14.16	3-Octanone *^2^	C_8_H_16_O	128.1	9.5	2.1	0.11
16.04	2,6-Dimethyl-2,4,6-Octatriene *^3^	C_10_H_16_	136.1	15	2.2	0.94
17.11	1-Octen-3-ol *^3^	C_8_H_16_O	128.1	1.1	1.0	0.25
18.50	Linalool *^1^ *^2^	C_10_H_18_O	154.1	27	41	22
18.61	Benzaldehyde *^3^	C_7_H_6_O	106.0	8.8	12	3.5
18.70	Linalyl acetate *^1^ *^2^	C_12_H_20_O_2_	196.1	9.6	17	8.4
19.35	Unknown *^3^	-	-	0.68	2.3	1.5
19.56	Terpinen-4-ol *^1^ *^2^	C_10_H_18_O	154.1	0.6	1.1	0.65
19.68	Caryophyllene *^3^	C_15_H_24_	204.2	0.43	1.0	0.71
20.89	Borneol *^3^	C_10_H_18_O	154.2	1.6	2.1	1.5
20.93	(+) or (-)-Lavandulol acetate *^2^	C_12_H_20_O_2_	196.1	1.6	1.7	0.91
21.30	(+) or (-)-Lavandulol acetate *^2^	C_12_H_20_O_2_	196.1	1.3	1.4	1.0

*^1^ Effective components, *^2^ Components known their compounding ratios in each essential oil, *^3^ Concentrations calculated based on the peak area of linalool, *^4^ Contaminant components in the room air, *^5^ Concentrations based on the calibration gas obtained by volatilizing 1 μL of essential oil in 10 L of air in a PDVF bag.

**Table 4 ijerph-19-02904-t004:** The concentration of each component from room air including diffused tea tree oil.

Retention Time [min]	Components	Molecular Formula	Molecular Weight	Concentration [ppt] *^6^		
				0 min	30 min	60 min
6.55	Ethyl Acetate *^3^ *^4^	C_4_H_8_O_2_	88.1	6.2	4.8	14
6.71	Methanol *^5^	CH_4_O	32.0	-	-	-
6.84	Methyl ethyl ketone *^3^	C_4_H_8_O	72.1	1.6	2.6	4.1
7.71	Ethanol *^3^ *^4^	C_2_H_6_O	46.0	20	32	25
7.81	Benzene *^3^ *^4^	C_6_H_6_	78.0	14	14	13
9.64	α-Pinene *^1^ *^2^	C_10_H_16_	136.1	25	0.41	0.19
11.32	β-Pinene *^1^ *^2^	C_10_H_16_	136.1	6.9	0.13	0.073
11.56	(+) or (-)-β-Phellandrene *^2^	C_10_H_16_	136.1	7.8	0.26	0.15
12.77	α-terpinene *^1^ *^2^	C_10_H_16_	136.1	54	0.55	0.38
13.14	D-Limonene *^1^ *^2^	C_10_H_16_	136.1	5.1	0.24	0.18
13.36	(+) or (-)-β-Phellandrene *^2^	C_10_H_16_	136.1	3.9	0.31	0.24
13.51	1,8-Cineole *^1^ *^2^	C_10_H_18_O	154.1	8.7	2.3	0.54
14.03	γ-terpinene *^1^ *^2^	C_10_H_16_	136.1	104	2.6	0.88
14.49	p-cymene *^1^ *^2^	C_10_H_14_	134.1	26	1.8	1.1
14.70	Terpinolene *^2^	C_10_H_16_	136.1	14	0.39	-
19.58	Terpinen-4-ol *^1^ *^2^	C_10_H_18_O	154.1	16	22	19
21.49	α-Terpinenol *^1^ *^5^	C_10_H_18_O	154.1	-	-	-

*^1^ 14 effective components, *^2^ Components known their compounding ratios in each essential oil, *^3^ Concentrations calculated based on the peak area of γ-terpinene, *^4^ Contaminant components in the room air, *^5^ Components detected only from the reference gas in the gas bag. *^6^ Concentrations based on the calibration gas obtained by volatilizing 1 μL of essential oil in 10 L of air in PDVF bag.

**Table 5 ijerph-19-02904-t005:** The concentration of each component from room air including diffused Eucalyptus oil.

Retention Time [min]	Components	Molecular Formula	Molecular Weight	Concentration [ppt] *^6^		
				0 min	30 min	60 min
6.56	Ethyl Acetate *^3^ *^4^	C_4_H_8_O_2_	88.1	1.9	0.93	0.80
7.73	Ethanol *^3^ *^4^	C_2_H_6_O	46.0	6.9	5.9	21
7.81	Benzene *^3^ *^4^	C_6_H_6_	78.0	3.9	4.6	3.3
9.64	α-Pinene *^1^ *^2^	C_10_H_16_	136.1	168	2.2	1.3
11.32	β-Pinene *^1^ *^3^	C_10_H_16_	136.1	8.7	0.46	-
12.77	α-terpinene *^1^ *^3^	C_10_H_16_	136.1	0.57	-	-
13.16	D-Limonene *^1^ *^2^	C_10_H_16_	136.1	49	0.69	-
13.66	1,8-Cineole *^1^ *^2^	C_10_H_18_O	154.1	402	105	32
14.06	γ-terpinene *^1^ *^3^	C_10_H_16_	136.1	10	-	-
14.51	p-cymene *^1^ *^3^	C_10_H_14_	134.1	14	0.30	0.10
19.57	Terpinen-4-ol *^1^ *^3^	C_10_H_18_O	154.1	-	-	-
21.49	α-Terpinenol *^1^ *^3^ *^5^	C_10_H_18_O	154.1	-	-	-

*^1^ 14 effective components, *^2^ Components known their compounding ratios in each essential oil, *^3^ Concentrations calculated based on the peak area of 1,8-cineole, *^4^ Contaminant components in the room air, *^5^ Components detected only from the reference gas in the gas bag. *^6^ Concentrations based on the calibration gas obtained by volatilizing 1 μL of essential oil in 10 L of air in PDVF bag.

**Table 6 ijerph-19-02904-t006:** The concentration of each component from room air including diffused melissa oil.

Retention Time [min]	Components	Molecular Formula	Molecular Weight	Concentration [ppt] *^6^		
				0 min	30 min	60 min
6.55	Ethyl Acetate *^3^ *^4^	C_4_H_8_O_2_	88.1	3.1	6.2	11
6.72	Methyl Alcohol *^5^	CH_4_O	32.0	-	-	-
6.83	Methyl ethyl ketone *^3^	C_4_H_8_O	72.1	3.8	3.4	5.1
7.72	Ethanol *^3^ *^4^	C_2_H_6_O	46.0	43	40	119
7.81	Benzene *^3^ *^4^	C_6_H_6_	78.0	27	22	22
8.57	n-Propyl acetate	C_5_H_10_O_2_	102.1	5.7	5.5	6.9
10.07	Toluene *^3^ *^4^	C_7_H_8_	92.1	22	19	25
11.75	Ethylbenzene *^3^ *^4^	C_8_H_10_	106.1	4.3	3.0	5.4
12.23	1-Butanol *^3^	C_4_H_10_O	74.1	1.3	2.1	2.0
12.30	Unknown *^5^	-	-	-	-	-
13.13	D-Limonene *^5^	C_10_H_16_	136.1	-	-	-
13.28	3-Methyl-2-Butenal *^5^	C_5_H_8_O	84.1	-	-	-
13.47	Unknown *^5^	-	-	-	-	-
13.99	3-Carene *^5^	C_10_H_16_	136.1	-	-	-
15.23	3-Methyl-2-buten-1-ol *^5^	C_5_H_10_O	86.1	-	-	-
17.20	Acetic acid *^5^	C_2_H_4_O_2_	60.0	-	-	-
17.71	Citronellal *^1^ *^2^	C_10_H_18_O	154.1	16	21	14
18.47	linalool *^1^ *^3^	C_12_H_20_O_2_	196.1	-	2.3	1.1
18.60	Benzaldehyde *^3^	C_7_H_6_O	106.0	2.5	2.6	3.1
19.68	Caryophyllene *^2^	C_15_H_24_	204.2	1.9	3.1	0.71
20.59	Neral *^1^ *^2^	C_10_H_16_O	152.1	-	14	4.2
21.20	Geranial *^1^ *^2^	C_10_H_16_O	152.1	-	17	6.6
21.35	Unknown *^3^	-	-	-	3.0	1.3
21.48	Unknown *^3^	-	-	-	2.4	1.1
21.85	Unknown *^3^	-	-	-	0.67	-
22.36	Unknown *^3^	-	-	1.2	7.4	2.8

* ^1^ 14 effective components, *^2^ Components known their compounding ratios in each essential oil, *^3^ Concentrations calculated based on the peak area of caryophyllene, *^4^ Contaminant components in the room air, *^5^ Components detected only from the reference gas in the gas bag. *^6^ Concentrations based on the calibration gas obtained by volatilizing 1 μL of essential oil in 10 L of air in PDVF bag.

**Table 7 ijerph-19-02904-t007:** Summary of GC/MS analyses to each diffused essential oil.

	Lavender	Tee Tree Oil	Eucalyptus	Melissa
Temporal change of total concentration	Constant level of 39–75 ppt for 60 min	272 ppt at start, and its 31 ppt after 30 min	555 ppt at start, its 100 ppt after 30 min	Constant level of 17–55 ppt for 60 min
Concentration decrease over time	Slight decrease	Explicit decrease	Explicit decrease	Slight decrease
concentration peak delay	Yes	No	No	Yes

## Data Availability

No applicable.

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
