# Peer review of "Examination of VOC Concentration of Aroma Essential Oils and Their Major VOCs Diffused in Room Air"

_ijerph, 2022, doi:10.3390/ijerph19052904_

Round 1

Reviewer 1 Report

Attached file

Author Response

Thank you for your valuable comments and advice. I have revised and added the parts of manuscript referring advice. Please see the attachment file where I would like to answer your comments in detail.

Reviewer 2 Report

The references could be improved.

Author Response

Thanks for kind advice and recommendation.

Reviewer 3 Report

Comments and Suggestions for Authors

Title ‎

It is recommended to revise the title and make it concise to the work.

Abstract

It is very short and need to be redesigned ‎with a ‎clear flow (background, aim, methods ‎used, findings, significance).‎

It is very less and need to be redesigned to give more about literature, to demonstrate the results and the importance of work.

For example,

Line 14: why did the authors select only these 14 constituents despite the oils contain other important constituents?

Line 15-19 it is one statement which is very long and confusing.

Introduction

Line29: better to replace publication with studies or reports.

Line33: better to replace composition with constituents.

 Line 33-36: this statement is long better to rephrase.

Line 38:  replace ingredient with ingredients as a plural

Line 48: it is noticed the authors use British English, so preferable to replace odor (US English) with odour (British English).

Line 51: add “s” to VOC to be VOCs

Line 54: ingredient should be plural “ingredients”

Line 54-59: it is long and confusing statement. Please, revise and rephrase it in a simple way.

In general, introduction needs more improvement especially regarding the references. It was notices from line 33 till line 59 there were no references to ensure the mentioned information.

Materials and methods

Line 61: specify the name of section “Subsection..)

Line 66- 71: internet commercial web information is not a convenient nor scientifically to retrieve information for scientific publication. So, it is crucial to retrieve and provide the references from well stablished scientific data base.

-Provide the full scientific name of plants from which oils were prepared.

Paragraph from Line 74 to line 71, needs to be redesigned and expressing why the authors did selection these 14 constituents.

Line 77 starts with abbreviated “fig. 1”. It is better to start with full word “figure 1”‎and make uniformity during usage of full and abbreviated words for the whole manuscript.

For the selected components in figure 1, better to write only one name for each and if it is important to mention the other names, provide it in the text itself.  

The authors provided the IUPAC names of components ‎ (10-14) only and some other names for components (5 and 6). Better to make uniformity and consistency of work. So, preferable to remove the IUPAC names.

Line 86-87:  better to make a uniformity of units, so, replace any” ml” with “mL”, the same in Line 97 “200 ml/min”. The same for minutes keep it to be “min” for the whole manuscript.

Section ‎2.2 Diffusion of essential oil and VOC into room air.

It was better to perform the experiments under the same every conditions” I mean all in draft chamber or office room” to give better comparison and reflect the precise diffusion for all tested conditions

Section 2.3 GC/MS:

This section needs references for the selected conditions.

What was the standard used to compare with.

The authors did the diffusion of the 14 selected components but they did not perform GC/MS analysis for such sample to compare with the selected oils. So, it preferable to do it.

Results

-Line 140: figure 6 shows the GC/MS chromatogram not spectra, please correct it in the whole manuscript.

-Figure 6 it is very confusing so no need to insert the structure of the constituents “name is enough”. Revise the title.

-It will be better and informative to compare with those of 14 selected constituents please insert its chromatogram “14 selected constituents”

- Better to add all oils don’t select lavender oil and keep the remaining oils as supplementary.

For tables 3, 4, 5 and 6 it is very essential to provide the retention index “RI” to demonstrate a solid confirmation of identity of constituents. Provide the molecular mass “MS” with four digits as it came from the results. Specify the meaning of ppt in table’s captions.

The titles of figures 7, 8, 9 and 10 replace “VOC” with “VOCs” and please consider this for the whole manuscript.  Specify the meaning of ppt in figures’ caption and the text at the first appearance.‎

Paragraph from line 182 to 191 need improvement to express the importance.

For figure 11, specify the meaning of ppb in figures’ caption and the text at the first appearance

Discussion

Discussion part needs improvement for the start of importance of work, follow of information and comparison with some pervious work to rational the impact of current study as a new additive information.

Line 252-256. Authors wrote three basic categories but what are these categories. These paragraph needs to be rephrased with better explanation with current work.

Table 7. Better to be shifted to results section and specify the meaning of ppt in table’s captions.

Provide full meanings of IFRA “Line 294” and RIFM “Line 296”.

Replace VOC with VOCs in lines; 284, 288, 289,290, 294, 300 and 301.

Provide references for lines 292-294.

Conclusion

Conclusion needs for better expression of importance of study. Avoid long sentences.   

Author Response

(The authors gave the same response as above.)

Round 2

Reviewer 1 Report

The authors have improved the manuscript. However, they should correct the added new references e.g 9, 13, 15, 17, 19, 20 (standards not met). The manuscript can be accepted after minor revision.

Author Response

Thank you for reviewers’ valuable comments. I have revised the new references according to the standards.

Reviewer 3 Report

The authors did most of comments but for improvement, please consider the following:

-"Minutes" keep it to be “min” for the whole manuscript. For example; Line: 96, 106, 107, 108 and 109

-Write full word figure 3a in line 104. consider the same for similar situations. 

-In “section 2.3. GC/MS analysis” and “section 3.1 GC/MS results of essential oils diffused into office room” please specify how could the authors identify the components and references for identification to confirm the identity?

-Provide the molecular mass “MS” with four digits as it came from the results

-Specify the meaning of ppt in table’s captions “table 3, 4, 5 and 6.

- Line 164 “The concentration of each component is calculated “change to be “The concentration of each component was calculated” for uniformity of tenses.

-Line 165 “For the components which was unknown, or chemicals originated”   change to be ‎ “For the components which were unknown, or chemicals originated”

-Line 275 change “reduce” to be “reduction”

-In general, it is better to revise the “English language”, especially plural and singular words and their associated verbs as well as tenses.

-In supplementary materials change the title from “GC/MS spectra” to “GC/MS chromatogram”

Author Response

>Thank you for reading our manuscript again and your kind advice. We have revised the manuscript following your comments.

-"Minutes" keep it to be “min” for the whole manuscript. For example; Line: 96, 106, 107, 108 and 109
-Write full word figure 3a in line 104. consider the same for similar situations.
>Thanks for valuable comments. I have replaced and revised the manuscript.

-In “section 2.3. GC/MS analysis” and “section 3.1 GC/MS results of essential oils diffused into office room” please specify how could the authors identify the components and references for identification to confirm the identity?
>Concentrations based on the calibration gas obtained by volatilizing 1 μL of representative feature component, linalool γ-terpinene, 1,8-cineole, and caryophyllene, for lavender, tea tree, eucalyptus, and melissa in 10 L of air in PDVF bag. I have added this information in the revised the manuscript.

-Provide the molecular mass “MS” with four digits as it came from the results
-Specify the meaning of ppt in table’s captions “table 3, 4, 5 and 6.
- Line 164 “The concentration of each component is calculated “change to be “The concentration of each component was calculated” for uniformity of tenses.
-Line 165 “For the components which was unknown, or chemicals originated”   change to be ‎ “For the components which were unknown, or chemicals originated”
-Line 275 change “reduce” to be “reduction”
-In general, it is better to revise the “English language”, especially plural and singular words and their associated verbs as well as tenses.

>Thanks for valuable advice. I have revised the points reviewer commented and checked again the manuscript carefully with the English use of plural and singular, tenses.

-In supplementary materials change the title from “GC/MS spectra” to “GC/MS chromatogram”

>Thanks for valuable comments. I have revised the captions of Figure of S1, S2, S3 placed and added the first page including the title of the manuscript.